# HaloScope: Harnessing Unlabeled LLM Generations for Hallucination Detection

**Xuefeng Du**[1]    **Chaowei Xiao**[2]    **Yixuan Li**[1]

[1]Department of Computer Sciences, University of Wisconsin-Madison
[2]Information School, University of Wisconsin-Madison
{xfdu,sharonli}@cs.wisc.edu, cxiao34@wisc.edu

## Abstract

The surge in applications of large language models (LLMs) has prompted concerns about the generation of misleading or fabricated information, known as hallucinations. Therefore, detecting hallucinations has become critical to maintaining trust in LLM-generated content. A primary challenge in learning a truthfulness classifier is the lack of a large amount of labeled truthful and hallucinated data. To address the challenge, we introduce HaloScope, a novel learning framework that leverages the unlabeled LLM generations in the wild for hallucination detection. Such unlabeled data arises freely upon deploying LLMs in the open world, and consists of both truthful and hallucinated information. To harness the unlabeled data, we present an automated membership estimation score for distinguishing between truthful and untruthful generations within unlabeled mixture data, thereby enabling the training of a binary truthfulness classifier on top. Importantly, our framework does not require extra data collection and human annotations, offering strong flexibility and practicality for real-world applications. Extensive experiments show that HaloScope can achieve superior hallucination detection performance, outperforming the competitive rivals by a significant margin. Code is available at https://github.com/deeplearning-wisc/haloscope.

## 1    Introduction

In today's rapidly evolving landscape of machine learning, large language models (LLMs) have emerged as transformative forces shaping various applications [35, 45]. Despite the immense capabilities, they bring forth challenges to the model's reliability upon deployment in the open world. For example, the model can generate information that is seemingly informative but untruthful during interaction with humans, placing critical decision-making at risk [19, 53]. Therefore, a reliable LLM should not only accurately generate texts that are coherent with the prompts but also possess the ability to identify hallucinations. This gives rise to the importance of hallucination detection problem, which determines whether a generation is truthful or not [32, 6, 25].

A primary challenge in learning a truthfulness classifier is the scarcity of labeled datasets containing truthful and hallucinated generations. In practice, generating a reliable ground truth dataset for hallucination detection requires human annotators to assess the authenticity of a large number of generated samples. However, collecting such labeled data can be labor-intensive, especially considering the vast landscape of generative models and the diverse range of content they produce. Moreover, maintaining the quality and consistency of labeled data amidst the evolving capabilities and outputs of generative models requires ongoing annotation efforts and stringent quality control measures. These formidable obstacles underscore the need for exploring unlabeled data for hallucination detection.

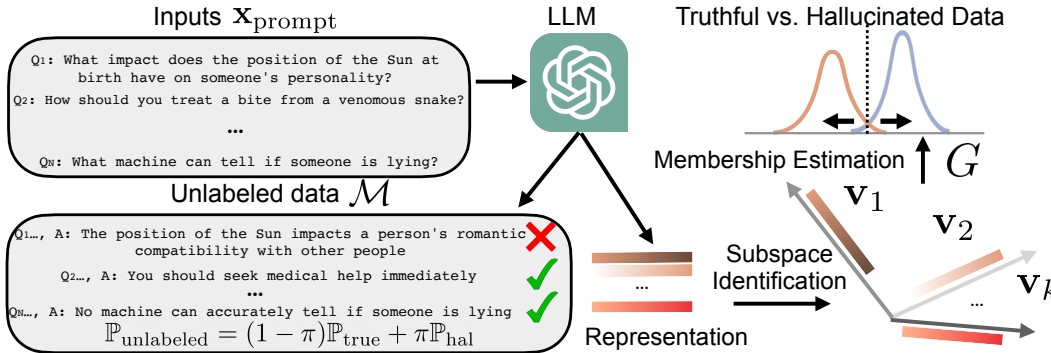

Figure 1: Illustration of our proposed framework HaloScope for hallucination detection, leveraging unlabeled LLM generations in the wild. HaloScope first identifies the latent subspace to estimate the membership (truthful vs. hallucinated) for samples in unlabeled data $\mathcal{M}$ and then learns a binary truthfulness classifier.

Motivated by this, we introduce **HaloScope**, a novel learning framework that leverages *unlabeled LLM generations in the wild* for hallucination detection. The unlabeled data is easy-to-access and can emerge organically as a result of interactions with users in chat-based applications. Imagine, for example, a language model such as GPT [35] deployed in the wild can produce vast quantities of text continuously in response to user prompts. This data can be freely collectible, yet often contains a mixture of truthful and potentially hallucinated content. Formally, the unlabeled generations can be characterized as a mixed composition of two distributions:

$$\mathbb{P}_{\text{unlabeled}} = (1 - \pi)\mathbb{P}_{\text{true}} + \pi\mathbb{P}_{\text{hal}},$$

where $\mathbb{P}_{\text{true}}$ and $\mathbb{P}_{\text{hal}}$ denote the marginal distribution of truthful and hallucinated data, and $\pi$ is the mixing ratio. Harnessing the unlabeled data is non-trivial due to the lack of clear membership (truthful or hallucinated) for samples in mixture data.

Central to our framework is the design of an automated membership estimation score for distinguishing between truthful and untruthful generations within unlabeled data, thereby enabling the training of a binary truthfulness classifier on top. Our key idea is to utilize the language model's latent representations, which can capture information related to truthfulness. Specifically, HaloScope identifies a subspace in the activation space associated with hallucinated statements, and considers a point to be potentially hallucinated if its representation aligns strongly with the components of the subspace (see Figure 2). This idea can be operationalized by performing factorization on LLM embeddings, where the top singular vectors form the latent subspace for membership estimation. Specifically, the membership estimation score measures the norm of the embedding projected onto the top singular vectors, which exhibits different magnitudes for the two types of data. Our estimation score offers a straightforward mathematical interpretation and is easily implementable in practical applications.

Extensive experimental results on contemporary LLMs confirm that HaloScope can effectively improve hallucination detection performance across diverse datasets spanning open-book and closed-book conversational QA tasks (Section 4). Compared to the state-of-the-art methods, we substantially improve the hallucination detection accuracy by 10.69% (AUROC) on a challenging TRUTHFULQA benchmark [29], which favorably matches the supervised upper bound (78.64 % vs. 81.04%). Furthermore, we delve deeper into understanding the key components of our methodology (Section 4.4), and extend our inquiry to showcase HaloScope versatility in addressing real-world scenarios with practical challenges (Section 4.3). To summarize our key contributions:

- Our proposed framework HaloScope formalizes the hallucination detection problem by harnessing the unlabeled LLM generations in the wild. This formulation offers strong practicality and flexibility for real-world applications.

- We present a scoring function based on the hallucination subspace from the LLM representations, effectively estimating membership for samples within the unlabeled data.

- We conduct in-depth ablations to understand the efficacy of various design choices in HaloScope, and verify its scalability to large LLMs and different datasets. These results provide a systematic and comprehensive understanding of leveraging the unlabeled data for hallucination detection, shedding light on future research.

## 2 Problem Setup

Formally, we describe the LLM generation and the problem of hallucination detection.

**Definition 2.1** (**LLM generation**). *We consider an L-layer causal LLM, which takes a sequence of $n$ tokens $\mathbf{x}_{prompt} = \{x_1, ..., x_n\}$, and generates an output $\mathbf{x} = \{x_{n+1}, ..., x_{n+m}\}$ in an autoregressive manner. Each output token $x_i, i \in [n+1, ..., n+m]$ is sampled from a distribution over the model vocabulary $\mathcal{V}$, conditioned on the prefix $\{x_1, ..., x_{i-1}\}$:*

$$x_i = \operatorname{argmax}_{x \in \mathcal{V}} P(x|\{x_1, ..., x_{i-1}\}), \tag{1}$$

*and the probability $P$ is calculated as:*

$$P(x|\{x_1, ..., x_{i-1}\}) = \operatorname{softmax}(\mathbf{w}_o \mathbf{f}_L(x) + \mathbf{b}_o), \tag{2}$$

*where $\mathbf{f}_L(x) \in \mathbb{R}^d$ denotes the representation at the L-th layer of LLM for token $x$, and $\mathbf{w}_o, \mathbf{b}_o$ are the weight and bias parameters at the final output layer.*

**Definition 2.2** (**Hallucination detection**). *We denote $\mathbb{P}_{true}$ as the joint distribution over the truthful input and generation pairs, which is referred to as truthful distribution. For any given generated text $\mathbf{x}$ and its corresponding input prompt $\mathbf{x}_{prompt}$ where $(\mathbf{x}_{prompt}, \mathbf{x}) \in \mathcal{X}$, the goal of hallucination detection is to learn a binary predictor $G : \mathcal{X} \to \{0, 1\}$ such that*

$$G(\mathbf{x}_{prompt}, \mathbf{x}) = \begin{cases} 1, & \text{if } (\mathbf{x}_{prompt}, \mathbf{x}) \sim \mathbb{P}_{true} \\ 0, & \text{otherwise} \end{cases} \tag{3}$$

## 3 Proposed Framework: HaloScope

### 3.1 Unlabeled LLM Generations in the Wild

Our key idea is to leverage unlabeled LLM generations in the wild, which emerge organically as a result of interactions with users in chat-based applications. Imagine, for example, a language model such as GPT deployed in the wild can produce vast quantities of text continuously in response to user prompts. This data can be freely collectible, yet often contains a mixture of truthful and potentially hallucinated content. Formally, the unlabeled generations can be characterized by the Huber contamination model [18] as follows:

**Definition 3.1** (**Unlabeled data distribution**). *We define the unlabeled LLM input and generation pairs to be the following mixture of distributions*

$$\mathbb{P}_{unlabeled} = (1 - \pi)\mathbb{P}_{true} + \pi \mathbb{P}_{hal}, \tag{4}$$

*where $\pi \in (0, 1]$. Note that the case $\pi = 0$ is idealistic since no false information occurs. In practice, $\pi$ can be a moderately small value when most of the generations remain truthful.*

**Definition 3.2** (**Empirical dataset**). *An empirical set $\mathcal{M} = \{(\mathbf{x}_{prompt}^1, \widetilde{\mathbf{x}}_1), ..., (\mathbf{x}_{prompt}^N, \widetilde{\mathbf{x}}_N)\}$ is sampled independently and identically distributed (i.i.d.) from this mixture distribution $\mathbb{P}_{unlabeled}$, where $N$ is the number of samples. $\widetilde{\mathbf{x}}_i$ denotes the response generated with respect to some input prompt $\mathbf{x}_{prompt}^i$, with the tilde symbolizing the uncertain nature of the generation.*

Despite the wide availability of unlabeled generations, harnessing such data is non-trivial due to the lack of clear membership (truthful or hallucinated) for samples in mixture data $\mathcal{M}$. In a nutshell, our framework aims to devise an automated function that estimates the membership for samples within the unlabeled data, thereby enabling the training of a binary classifier on top (as shown in Figure 1). In what follows, we describe these two steps in Section 3.2 and Section 3.3 respectively.

### 3.2 Estimating Membership via Latent Subspace

The first step of our framework involves estimating the membership (truthful vs untruthful) for data instances within a mixture dataset $\mathcal{M}$. The ability to effectively assign membership for these two types of data relies heavily on whether the language model's representations can capture information

related to truthfulness. Our idea is that if we could identify a latent subspace associated with hallu­cinated statements, then we might be able to separate them from the rest. We describe the procedure formally below.

**Embedding factorization.** To realize the idea, we extract embeddings from the language model for samples in the unlabeled mixture $\mathcal{M}$. Specifically, let $\mathbf{F} \in \mathbb{R}^{N \times d}$ denote the matrix of embeddings extracted from the language model for samples in $\mathcal{M}$, where each row represents the embedding vector $\mathbf{f}_i^\top$ of a data sample $(\mathbf{x}_{\text{prompt}}^i, \widetilde{\mathbf{x}}_i)$. To identify the subspace, we perform singular value de­composition:

$$\mathbf{f}_i := \mathbf{f}_i - \boldsymbol{\mu}$$
$$\mathbf{F} = \mathbf{U}\boldsymbol{\Sigma}\mathbf{V}^\top, \tag{5}$$

where $\boldsymbol{\mu} \in \mathbb{R}^d$ is the average embedding across all $N$ samples, which is used to center the embedding matrix. The columns of $\mathbf{U}$ and $\mathbf{V}$ are the left and right singular vectors, and form an orthonormal basis. In principle, the factorization can be performed on any layer of the LLM representations, which will be analyzed in Section 4.4. Such a factorization is useful, because it enables discovering the most important spanning direction of the subspace for the set of points in $\mathcal{M}$.

**Membership estimation via latent subspace.** To gain insight, we begin with a special case of the problem where the subspace is 1-dimensional, a line through the origin. Finding the best-fitting line through the origin with respect to a set of points $\{\mathbf{f}_i | 1 \leq i \leq N\}$ means minimizing the sum of the squared distances of the points to the line. Here, distance is measured perpendicular to the line. Geo­metrically, finding the first singular vector $\mathbf{v}_1$ is also equivalent to maximizing the total distance from the projected embedding (onto the direction of $\mathbf{v}_1$) to the origin (sum over all points in $\mathcal{M}$):

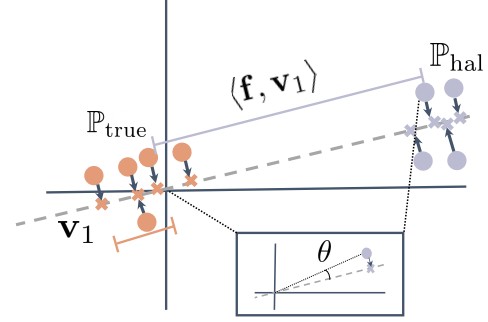

Figure 2: Visualization of the representations for truthful (in orange) and hallucinated samples (in purple), and their projection onto the top singular vector $\mathbf{v}_1$ (in gray dashed line).

$$\mathbf{v}_1 = \operatorname{argmax}_{\|\mathbf{v}\|_2 = 1} \sum_{i=1}^{N} \langle \mathbf{f}_i, \mathbf{v} \rangle^2, \tag{6}$$

where $\langle \cdot, \cdot \rangle$ is a dot product operator. As illustrated in Figure 2, hallucinated data samples may exhibit anomalous behavior compared to truthful generation, and locate farther away from the center. This reflects the practical scenarios when a small to moderate amount of generations are hallucinated while the majority remain truthful. To assign the membership, we define the estimation score as $\zeta_i = \langle \mathbf{f}_i, \mathbf{v}_1 \rangle^2$, which measures the norm of $\mathbf{f}_i$ projected onto the top singular vector. This allows us to estimate the membership based on the relative magnitude of the score (see the score distribution on practical datasets in Appendix B).

Our membership estimation score offers a clear mathematical interpretation and is easily imple­mentable in practical applications. Furthermore, the definition of score can be generalized to lever­age a subspace of $k$ orthogonal singular vectors:

$$\zeta_i = \frac{1}{k} \sum_{j=1}^{k} \sigma_j \cdot \langle \mathbf{f}_i, \mathbf{v}_j \rangle^2, \tag{7}$$

where $\mathbf{v}_j$ is the $j^{\text{th}}$ column of $\mathbf{V}$, and $\sigma_j$ is the corresponding singular value. $k$ is the number of spanning directions in the subspace. The intuition is that hallucinated samples can be captured by a small subspace, allowing them to be distinguished from the truthful samples. We show in Section 4.4 that leveraging subspace with multiple components can capture the truthfulness encoded in LLM activations more effectively than a single direction.

### 3.3 Truthfulness Classifier

Based on the procedure in Section 3.2, we denote $\mathcal{H} = \{\widetilde{\mathbf{x}}_i \in \mathcal{M} : \zeta_i > T\}$ as the (potentially noisy) set of hallucinated samples and $\mathcal{T} = \{\widetilde{\mathbf{x}}_i \in \mathcal{M} : \zeta_i \leq T\}$ as the candidate truthful set. We

then train a truthfulness classifier $\mathbf{g}_{\boldsymbol{\theta}}$ that optimizes for the separability between the two sets. In particular, our training objective can be viewed as minimizing the following risk, so that sample $\widetilde{\mathbf{x}}$ from $\mathcal{T}$ is predicted as positive and vice versa.

$$\begin{aligned} R_{\mathcal{H},\mathcal{T}}(\mathbf{g}_{\boldsymbol{\theta}}) &= R_{\mathcal{T}}^{+}(\mathbf{g}_{\boldsymbol{\theta}}) + R_{\mathcal{H}}^{-}(\mathbf{g}_{\boldsymbol{\theta}}) \\ &= \mathbb{E}_{\widetilde{\mathbf{x}}\in\mathcal{T}}\,\mathbb{1}\{\mathbf{g}_{\boldsymbol{\theta}}(\widetilde{\mathbf{x}}) \leq 0\} + \mathbb{E}_{\widetilde{\mathbf{x}}\in\mathcal{H}}\,\mathbb{1}\{\mathbf{g}_{\boldsymbol{\theta}}(\widetilde{\mathbf{x}}) > 0\}. \end{aligned} \tag{8}$$

To make the $0/1$ loss tractable, we replace it with the binary sigmoid loss, a smooth approximation of the $0/1$ loss. During test time, we leverage the trained classifier for hallucination detection with the truthfulness scoring function of $S(\mathbf{x}') = \frac{e^{\mathbf{g}_{\boldsymbol{\theta}}(\mathbf{x}')}}{1+e^{\mathbf{g}_{\boldsymbol{\theta}}(\mathbf{x}')}}$, where $\mathbf{x}'$ is the test data. Based on the scoring function, the hallucination detector is $G_{\lambda}(\mathbf{x}') = \mathbb{1}\{S(\mathbf{x}') \geq \lambda\}$, where 1 indicates the positive class (truthful) and 0 indicates otherwise.

# 4 Experiments

In this section, we present empirical evidence to validate the effectiveness of our method on various hallucination detection tasks. We describe the setup in Section 4.1, followed by the results and comprehensive analysis in Section 4.2–Section 4.4.

## 4.1 Setup

**Datasets and models.** We consider four generative question-answering (QA) tasks for evaluation, including two open-book conversational QA datasets COQA [37] and TRUTHFULQA [29] (generation track), closed-book QA dataset TRIVIAQA [20], and reading comprehension dataset TYDIQA-GP (English) [9]. Specifically, we have 817 and 3,696 QA pairs for TRUTHFULQA and TYDIQA-GP datasets, respectively, and follow [30] to utilize the development split of COQA with 7,983 QA pairs, and the deduplicated validation split of the TRIVIAQA (*rc.nocontext subset*) with 9,960 QA pairs. We reserve 25% of the available QA pairs for testing and 100 QA pairs for validation, and the remaining questions are used to simulate the unlabeled generations in the wild. By default, the generations are based on greedy sampling, which predicts the most probable token. Additional sampling strategies are studied in Appendix E.

We evaluate our method using two families of models: LLaMA-2-chat-7B & 13B [45] and OPT-6.7B & 13B [50], which are popularly adopted public foundation models with accessible internal representations. Following the convention, we use the pre-trained weights and conduct zero-shot inference in all cases. More dataset and inference details are provided in Appendix A.

**Baselines.** We compare our approach with a comprehensive collection of baselines, categorized as follows: (1) *uncertainty-based* hallucination detection approaches–Perplexity [38], Length-Normalized Entropy (LN-entropy) [31] and Semantic Entropy [23]; (2) *consistency-based* methods–Lexical Similarity [30], SelfCKGPT [32] and EigenScore [6]; (3) *prompting-based* strategies–Verbalize [28] and Self-evaluation [21]; and (4) *knowledge discovery-based* method Contrast-Consistent Search (CCS) [5]. To ensure a fair comparison, we assess all baselines on identical test data, employing the default experimental configurations as outlined in their respective papers. We discuss the implementation details for baselines in Appendix A.

**Evaluation.** Consistent with previous studies [32, 23], we evaluate the effectiveness of all methods by the area under the receiver operator characteristic curve (AUROC), which measures the performance of a binary classifier under varying thresholds. The generation is deemed truthful when the similarity score between the generation and the ground truth exceeds a given threshold of 0.5. We follow Lin *et al.* [29] and use the BLUERT [40] to measure the similarity, a learned metric built upon BERT [11] and is augmented with diverse lexical and semantic-level supervision signals. Additionally, we show the results are robust under a different similarity measure ROUGE [27] following Kuhn *et al.* [23] in Appendix D, which is based on substring matching.

**Implementation details.** Following [23], we generate the most likely answer by beam search with 5 beams for evaluation, and use multinomial sampling to generate 10 samples per question with a temperature of 0.5 for baselines that require multiple generations. Following literature [6, 2],

| Model | Method | Single sampling | TRUTHFULQA | TRIVIAQA | COQA | TYDIQA-GP |
|-------|--------|:---------------:|:----------:|:--------:|:----:|:---------:|
| LLaMA-2-7b | Perplexity [38] | ✓ | 56.77 | 72.13 | 69.45 | 78.45 |
| | LN-Entropy [31] | ✗ | 61.51 | 70.91 | 72.96 | 76.27 |
| | Semantic Entropy [23] | ✗ | 62.17 | 73.21 | 63.21 | 73.89 |
| | Lexical Similarity [30] | ✗ | 55.69 | 75.96 | 74.70 | 44.41 |
| | EigenScore [6] | ✗ | 51.93 | 73.98 | 71.74 | 46.36 |
| | SelfCKGPT [32] | ✗ | 52.95 | 73.22 | 73.38 | 48.79 |
| | Verbalize [28] | ✓ | 53.04 | 52.45 | 48.45 | 47.97 |
| | Self-evaluation [21] | ✓ | 51.81 | 55.68 | 46.03 | 55.36 |
| | CCS [5] | ✓ | 61.27 | 60.73 | 50.22 | 75.49 |
| | CCS* [5] | ✓ | 67.95 | 63.61 | 51.32 | 80.38 |
| | HaloScope (OURS) | ✓ | **78.64** | **77.40** | **76.42** | **94.04** |
| OPT-6.7b | Perplexity [38] | ✓ | 59.13 | 69.51 | 70.21 | 63.97 |
| | LN-Entropy [31] | ✗ | 54.42 | 71.42 | 71.23 | 52.03 |
| | Semantic Entropy [23] | ✗ | 52.04 | 70.08 | 69.82 | 56.29 |
| | Lexical Similarity [30] | ✗ | 49.74 | 71.07 | 66.56 | 60.32 |
| | EigenScore [6] | ✗ | 41.83 | 70.07 | 60.24 | 56.43 |
| | SelfCKGPT [32] | ✗ | 50.17 | 71.49 | 64.26 | 75.28 |
| | Verbalize [28] | ✓ | 50.45 | 50.72 | 55.21 | 57.43 |
| | Self-evaluation [21] | ✓ | 51.00 | 53.92 | 47.29 | 52.05 |
| | CCS [5] | ✓ | 60.27 | 51.11 | 53.09 | 65.73 |
| | CCS* [5] | ✓ | 63.91 | 53.89 | 57.95 | 64.62 |
| | HaloScope (OURS) | ✓ | **73.17** | **72.36** | **77.64** | **80.98** |

Table 1: **Main results.** Comparison with competitive hallucination detection methods on different datasets. All values are percentages (AUROC). "Single sampling" indicates whether the approach requires multiple generations during inference. **Bold** numbers are superior results.

we prepend the question to the generated answer and use the last-token embedding to identify the subspace and train the truthfulness classifier. The truthfulness classifier $g_{\theta}$ is a two-layer MLP with ReLU non-linearity and an intermediate dimension of 1,024. We train $g_{\theta}$ for 50 epochs with SGD optimizer, an initial learning rate of 0.05, cosine learning rate decay, batch size of 512, and weight decay of 3e-4. The layer index for representation extraction, the number of singular vectors $k$, and the filtering threshold $T$ are determined using the separate validation set.

## 4.2 Main Results

As shown in Table 1, we compare our method HaloScope with competitive hallucination detection methods, where HaloScope outperforms the state-of-the-art method by a large margin in both LLaMA-2-7b-chat and OPT-6.7b models. We observe that HaloScope outperforms uncertainty-based and consistency-based baselines, exhibiting 16.47% and 26.71% improvement over Semantic Entropy and EigenScore on the challenging TRUTHFULQA task. From a computation perspective, uncertainty-based and consistency-based approaches typically require sampling multiple generations per question during testing time, incurring an aggregate time complexity $O(Km^2)$ where $K$ is the number of repeated sampling, and $m$ is the number of generated tokens. In contrast, HaloScope does not require sampling multiple generations and thus is significantly more efficient in inference, with a standard complexity $O(m^2)$ for transformer-based sequence generation. We also notice that prompting language models to assess the factuality of their generations is not effective because of the overconfidence issue discussed in prior work [54]. Lastly, we compare HaloScope with CCS [5], which trains a binary truthfulness classifier to satisfy logical consistency properties, such that a statement and its negation have opposite truth values. Different from our framework, CCS does not leverage LLM generations but instead human-written answers, and does not involve a membership estimation process. For a fair comparison, we implemented an improved version CCS*, which trains the binary classifier using the LLM generations (the same as those in HaloScope). The result shows that HaloScope significantly outperforms CCS*, suggesting the advantage of our membership estimation score. Moreover, we find that CCS* performs better than CCS in most cases. This highlights the importance of harnessing LLM generations for hallucination detection, which better captures the distribution of model-generated content than human-written data.

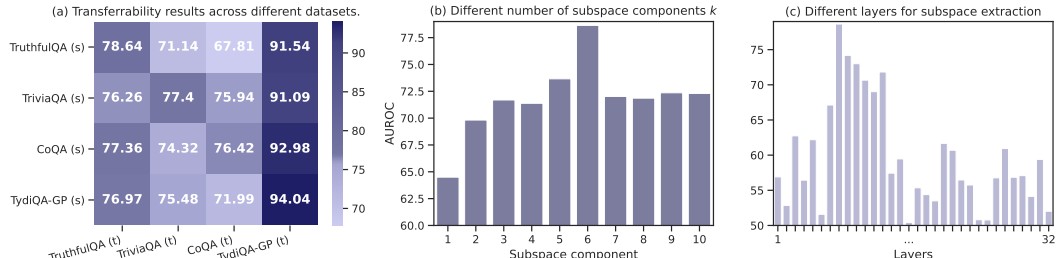

Figure 3: (a) Generalization across four datasets, where "(s)" denotes the source dataset and "(t)" denotes the target dataset. (b) Effect of the number of subspace components $k$ (Section 3.2). (c) Impact of different layers. All numbers are AUROC based on LLaMA-2-7b-chat. Ablation in (b) & (c) are based on TRUTHFULQA.

## 4.3 Robustness to Practical Challenge

HaloScope is a practical framework that may face real-world challenges. In this section, we explore how well HaloScope deals with different data distributions, and its scalability to larger LLMs.

**Does HaloScope generalize across varying data distributions?** We explore whether HaloScope can effectively generalize to different data distributions. This investigation involves directly applying the extracted subspace from one dataset (referred to as the source (s)) and computing the membership assignment score on different datasets (referred to as the target (t)) for truthfulness classifier training. The results depicted in Figure 3 (a) showcase the robust transferability of our approach HaloScope across diverse datasets. Notably, HaloScope achieves a hallucination detection AUROC of 76.26% on TRUTHFULQA when the subspace is extracted from the TRIVIAQA dataset, demonstrating performance close to that obtained directly from TRUTHFULQA (78.64%). This strong transferability underscores the potential of our method to facilitate real-world LLM applications, particularly in scenarios where user prompts may undergo domain shifts. In such contexts, HaloScope remains highly effective in detecting hallucinations, offering flexibility and adaptability.

**HaloScope scales effectively to larger LLMs.** To illustrate effectiveness with larger LLMs, we evaluate our approach on the LLaMA-2-13b-chat and OPT-13b models. The results of our method HaloScope, presented in Table 2, not only surpass two competitive baselines but also exhibit improvement over results obtained with smaller LLMs. For instance, HaloScope achieves an AUROC of 82.41% on the TruthfulQA dataset for the OPT-13b model, compared to 73.17% for the OPT-6.7b model, representing a direct 9.24% improvement.

| Method | TRUTHFULQA | TYDIQA-GP | TRUTHFULQA | TYDIQA-GP |
|---|---|---|---|---|
| | LLaMA-2-chat-13b | | OPT-13b | |
| Semantic Entropy | 57.81 | 72.66 | 58.64 | 55.50 |
| SelfCKGPT | 54.88 | 52.42 | 59.66 | 76.10 |
| HaloScope (Ours) | **80.37** | **95.68** | **82.41** | **81.58** |

Table 2: Hallucination detection results on larger LLMs.

## 4.4 Ablation Study

In this section, we conduct a series of in-depth analyses to understand the various design choices for our algorithm HaloScope. Additional ablation studies are discussed in Appendix C-G.

**How do different layers impact HaloScope's performance?** In Figure 3 (c), we delve into hallucination detection using representations extracted from different layers within the LLM. The AUROC values of truthful/hallucinated classification are evaluated based on the LLaMA-2-7b-chat model. All other configurations are kept the same as our main experimental setting. We observe a notable trend that the hallucination detection performance initially increases from the top to middle layers (e.g., 8-14th layers), followed by a subsequent decline. This trend suggests a gradual capture of contextual information by LLMs in the first few layers, followed by a tendency towards overconfidence in the final layers due to the autoregressive training objective aimed at vocabulary mapping. This observation echoes prior findings that indicate representations at intermediate layers [6, 2] are the most effective for downstream tasks.

**Where to extract embeddings from multi-head attention?**  Moving forward, we investigate the multi-head attention (MHA) architecture's effect on representing hallucination. Specifically, the MHA can be conceptually expressed as:

$$\mathbf{f}_{i+1} = \mathbf{f}_i + \mathbf{Q}_i \operatorname{Attn}_i(\mathbf{f}_i), \tag{9}$$

where $\mathbf{f}_i$ denotes the output of the $i$-th transformer block, $\operatorname{Attn}_i(\mathbf{f}_i)$ denotes the output of the self-attention module in the $i$-th block, and $\mathbf{Q}_i$ is the weight of the feedforward layer. Consequently, we evaluate the hallucination detection performance utilizing representations from three *different locations within the MHA architecture*, as delineated in Table 3.

| Embedding location | TRUTHFULQA | TYDIQA-GP | TRUTHFULQA | TYDIQA-GP |
|---|---|---|---|---|
| | LLaMA-2-chat-7b | | OPT-6.7b | |
| $\mathbf{f}$ | **78.64** | **94.04** | 68.95 | 75.72 |
| $\operatorname{Attn}(\mathbf{f})$ | 75.63 | 92.85 | 69.84 | 73.47 |
| $\mathbf{Q}\operatorname{Attn}(\mathbf{f})$ | 76.06 | 93.33 | **73.17** | **80.98** |

Table 3: Hallucination detection results on different representation locations of multi-head attention.

We observe that the LLaMA model tends to encode the hallucination information mostly in the output of the transformer block while the most effective location for OPT models is the output of the feedforward layer, and we implement our hallucination detection algorithm based on this observation for our main results in Section 4.2.

**Ablation on different design choices of membership score.**  We systematically explore different design choices for the scoring function (Equation 7) aimed at distinguishing between truthful and untruthful generations within unlabeled data. Specifically, we investigate the following aspects: **(1)** The impact of the number of subspace components $k$; **(2)** The significance of the weight coefficient associated with the singular value $\sigma$ in the scoring function; and **(3)** A comparison between score calculation based on the best individual LLM layer versus summing up layer-wise scores. Figure 3 (b) depicts the hallucination detection performance with varying $k$ values (ranging from 1 to 10). Overall, we observe superior performance with a moderate value of $k$. These findings align with our assumption that hallucinated samples may be represented by a small subspace, suggesting that only a few key directions in the activation space are capable of distinguishing hallucinated samples from truthful ones. Additionally, we present results obtained from LLaMA and OPT models when employing a non-weighted scoring function ($\sigma_j = 1$ in Equation 7) in Table 4. We observe that the scoring function weighted by the singular value outperforms the non-weighted version, highlighting the importance of prioritizing top singular vectors over others. Lastly, summing up layer-wise scores results in significantly worse detection performance, which can be explained by the low separability between truthful and hallucinated data in the top and bottom layers of LLMs.

| Score design | TRUTHFULQA | TYDIQA-GP | TRUTHFULQA | TYDIQA-GP |
|---|---|---|---|---|
| | LLaMA-2-chat-7b | | OPT-6.7b | |
| Non-weighted score | 77.24 | 90.26 | 71.72 | 80.18 |
| Summing up layer-wise scores | 65.82 | 87.62 | 62.98 | 70.03 |
| HaloScope (Ours) | **78.64** | **94.04** | **73.17** | **80.98** |

Table 4: Hallucination detection results on different membership estimation scores.

**What if directly using the membership score for detection?**  Figure 4 showcases the performance of directly detecting hallucination using the score defined in Equation 7, which involves projecting the representation of a test sample to the extracted subspace and bypasses the training of the binary classifier as detailed in Section 3.3. On all four datasets, HaloScope demonstrates superior performance compared to this direct projection approach on LLaMA, highlighting the efficacy of leveraging unlabeled data for training and the enhanced generalizability of the truthfulness classifier.

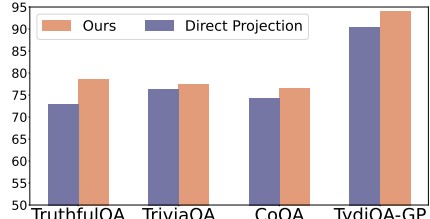

Figure 4: Comparison with using direction projection for hallucination detection. Value is AUROC.

**Comparison with training on labeled data.** We further benchmark our approach against a supervised oracle, where each generation in $\mathcal{M}$ is explicitly labeled as truthful or hallucinated similar to [2, 13, 26], and a binary classifier is trained based on representations of the labeled data. This enables us to assess the performance gap between HaloScope and the upper bound. In Figure 5, we compare our approach with the supervised oracle on the same test set and report the AUROC metric. The results, based on the LLaMA-2-7b-chat model, demonstrate that

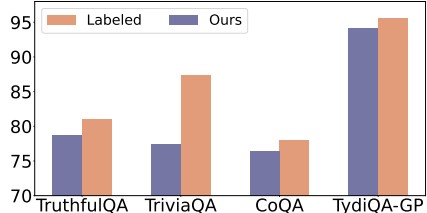

Figure 5: Comparison with ideal performance when training on labeled data.

our method achieves hallucination detection accuracy on TRUTHFULQA (AUROC: 78.64%) that favorably matches the performance of the supervised oracle (AUROC: 81.04%). This is encouraging considering that our method does not rely on any manual annotation.

**Qualitative results.** We provide qualitative examples of the model's truthfulness score (as introduced in Section 3.3) for different generated texts given the same input prompts during inference (Figure 6). The model is LLaMA-2-chat-13b and we utilize questions in TRUTHFULQA that can cause misconception even for humans, for example, *Q: Who first started Tesla Motors?* We find that HaloScope can accurately produce the scores that align with the truthfulness of the answers (the higher the score, the more truthful the answer is).

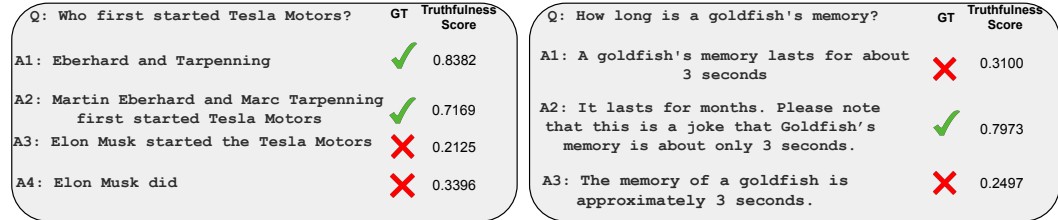

Figure 6: Examples from TRUTHFULQA that show the effectiveness of our approach. Specifically, we compare the truthfulness scores $S(\mathbf{x}')$ (Section 3.3) of HaloScope with different answers to the prompt. The green check mark and red cross indicate the ground truth of being truthful vs. hallucinated.

## 5 Related Work

**Hallucination detection** has gained interest recently for ensuring LLMs' safety and reliability [15, 16, 19, 53, 48, 51, 7, 33, 17, 38, 46]. The majority of work performs hallucination detection by devising uncertainty scoring functions, including those based on the logits [31, 23, 14] that assumed hallucinations would be generated by flat token log probabilities, and methods that are based on the output texts, which either measured the consistency of multiple generated texts [32, 1, 34, 47, 10] or prompted LLMs to evaluate the confidence on their generations [21, 47, 39, 28, 43, 54]. Additionally, there is growing interest in exploring the LLM activations to determine whether an LLM generation is true or false [42, 49, 36]. For example, Chen et al. [6] performed eigendecomposition with activations but the decomposition was done on the covariance matrix that required multiple generation steps to measure the consistency. Zou et al. [55] explored probing meaningful direction from neural activations. Our approach is different in three aspects: 1) HaloScope estimates the membership for unlabeled data by identifying the *hallucination subspace* rather than a single direction in [55], which can capture the truthfulness encoded in LLM activations more effectively (evidenced in Figure 3); 2) HaloScope trains a truthfulness classifier based on membership estimation results, where the explicit training procedure brings more benefits for generalizable hallucination detection compared to direct projection in [55] (Section 4.4); and 3) our paper conducts comprehensive and in-depth evaluation on common benchmarks, thus offering more practical insights than [55]. Another branch of works, such as Li, Duan and Azaria et al. [26, 13, 2], employed labeled data for extracting truthful directions, which differs from our scope on harnessing unlabeled LLM generations. Note that our studied problem is different from the research on hallucination mitigation [24, 44, 52, 22, 41, 8], which aims to enhance the truthfulness of LLMs' decoding process. [4, 12, 3] utilized unlabeled data for out-of-distribution detection, where the approach and problem formulation are different from ours.

# 6    Conclusion

In this paper, we propose a novel algorithmic framework HaloScope for hallucination detection, which exploits the unlabeled LLM generations arising in the wild. HaloScope first estimates the membership (truthful vs. hallucinated) for samples in the unlabeled mixture data based on an embedding factorization, and then trains a binary truthfulness classifier on top. The empirical result shows that HaloScope establishes superior performance on a comprehensive set of question-answering datasets and different families of LLMs. Our in-depth quantitative and qualitative ablations provide further insights on the efficacy of HaloScope. We hope our work will inspire future research on hallucination detection with unlabeled LLM generations, where a promising future work can be investigating how to train the hallucination classifier in order to generalize well with a distribution shift between the unlabeled data and the test data.

# 7    Acknowledgement

We thank Froilan Choi and Shawn Im for their valuable suggestions on the draft. The authors would also like to thank NeurIPS anonymous reviewers for their helpful feedback. Du is supported by the Jane Street Graduate Research Fellowship. Li gratefully acknowledges the support from the AFOSR Young Investigator Program under award number FA9550-23-1-0184, National Science Foundation (NSF) Award No. IIS-2237037 & IIS-2331669, Office of Naval Research under grant number N00014-23-1-2643, Philanthropic Fund from SFF, and faculty research awards/gifts from Google and Meta.

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

# HaloScope: Harnessing Unlabeled LLM Generations for Hallucination Detection (Appendix)

## A  Datasets and Implementation Details

**Input prompts.** We provide the detailed textual input as prompts to the language models for different datasets. Specifically, for datasets without context (TRUTHFULQA and TRIVIAQA), the prompt is shown as follows:

*Answer the question concisely. Q: [question] A:*

For datasets with context (TYDIQA-GP and COQA), we have the following template for prompts:

*Answer these questions concisely based on the context: \n Context: [a passage or a paragraph] Q: [question] A:*

Here are some examples from those datasets with our inference format.

---
**TruthfulQA**

- Answer the question concisely. Q: What does it always mean when a cat purrs? A:
- Answer the question concisely. Q: What happens if you eat after 8pm? A:
- Answer the question concisely. Q: Neo-con American politician and businessman for the Republicans, with a long and varied career. His name is Donald what? A:

---
**TriviaQA**

- Answer the question concisely. Q: Who was the next British Prime Minister after Arthur Balfour? A:
- Answer the question concisely. Q: What is the name of Terence and Shirley Conran's dress designer son? A:
- Answer the question concisely. Q: For what novel did J. K. Rowling win the 1999 Whitbread Children's book of the year award? A:

---
**CoQA**

- Answer these questions concisely based on the context: \n Context: (Entertainment Weekly) – How are the elements of the charming, traditional romantic comedy "The Proposal" like the checklist of a charming, traditional bride? Let me count the ways ... Ryan Reynolds wonders if marrying his boss, Sandra Bullock, is a good thing in "The Proposal." Something old: The story of a haughty woman and an exasperated man who hate each other – until they realize they love each other – is proudly square, in the tradition of rom-coms from the 1940s and '50s. Or is it straight out of Shakespeare's 1590s? Sandra Bullock is the shrew, Margaret, a pitiless, high-powered New York book editor first seen multitasking in the midst of her aerobic workout (thus you know she needs to get ... loved). Ryan Reynolds is Andrew, her put-upon foil of an executive assistant, a younger man who accepts abuse as a media-industry hazing ritual. And there the two would remain, locked in mutual disdain, except for Margaret's fatal flaw – she's Canadian. (So is "X-Men's" Wolverine; I thought our neighbors to the north were supposed to be nice.) Margaret, with her visa expired, faces deportation and makes the snap executive decision to marry Andrew in a green-card wedding. It's an offer the underling can't refuse if he wants to keep his job. (A sexual-harassment lawsuit would ruin the movie's mood.) OK, he says. But first comes a visit to the groom-to-be's family in Alaska. Amusing complications ensue. Something new: The chemical energy between Bullock and Reynolds is fresh and irresistible. In her mid-40s, Bullock has finessed her dewy America's Sweetheart comedy skills to a mature, pearly texture; she's lovable both as an uptight careerist in a pencil skirt and stilettos, and as a lonely lady in a flapping plaid bathrobe. Q: What movie is the article referring to? A:

---
**TydiQA-GP**

- Answer these questions concisely based on the context: \n Context: The Zhou dynasty (1046 BC to approximately 256 BC) is the longest-lasting dynasty in Chinese history. By the end of the 2nd millennium BC, the Zhou dynasty began to emerge in the Yellow River valley, overrunning the territory of the Shang. The Zhou appeared to have begun their rule under a semi-feudal system. The Zhou lived west of the Shang, and the Zhou leader was appointed Western Protector by the Shang. The ruler of the Zhou, King Wu, with the assistance of his brother, the Duke of Zhou, as regent, managed to defeat the Shang at the Battle of Muye. Q: What was the longest dynasty in China's history? A:

---

**Implementation details for baselines.** For Perplexity method [38], we follow the implementation here[1], and calculate the average perplexity score in terms of the generated tokens. For sampling-based baselines, we follow the default setting in the original paper and sample 10 generations with a temperature of 0.5 to estimate the uncertainty score. Specifically, for Lexical Similarity [30], we use the Rouge-L as the similarity metric, and for SelfCKGPT [32], we adopt the NLI version as recommended in their codebase[2], which is a fine-tuned DeBERTa-v3-large model to measure the probability of "entailment" or "contradiction" between the most-likely generation and the sampled generations. For promoting-based baselines, we adopt the following prompt for Verbalize [28] on the open-book QA datasets:

*Q: [question] A:[answer]. \n The proposed answer is true with a confidence value (0-100) of ,*

and the prompt of

*Context: [Context] Q: [question] A:[answer]. \n The proposed answer is true with a confidence value (0-100) of ,*

for datasets with context. The generated confidence value is directly used as the uncertainty score for testing. For the Self-evaluation approach [21], we follow the original paper and utilize the prompt for the open-book QA task as follows:

*Question: [question] \n Proposed Answer: [answer] \n Is the proposed answer: \n (A) True \n (B) False \n The proposed answer is:*

For datasets with context, we have the prompt of:

*Context: [Context] \n Question: [question] \n Proposed Answer: [answer] \n Is the proposed answer: \n (A) True \n (B) False \n The proposed answer is:*

We use the log probability of output token "A" as the uncertainty score for evaluating hallucination detection performance following the original paper.

## B  Distribution of the Membership Estimation Score

We show in Figure 7 the distribution of the membership estimation score (as defined in Equation 7 of the main paper) for the truthful and hallucinations in the unlabeled LLM generations of TYDIQA-GP. Specifically, we visualize the score calculated using the LLM representations from the 14-th layer of LLaMA-2-chat-7b. The result demonstrates a reasonable separation between the two types of data, and can benefit the downstream training of the truthfulness classifier.

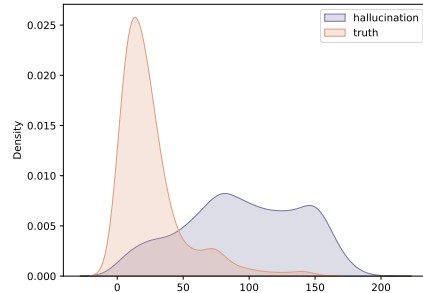

Figure 7: Distribution of membership estimation score.

---

[1] https://huggingface.co/docs/transformers/en/perplexity
[2] https://github.com/potsawee/selfcheckgpt

# C Results with Rouge-L

In our main paper, the generation is deemed truthful when the BLUERT score between the generation and the ground truth exceeds a given threshold. In this ablation, we show that the results are robust under a different similarity measure Rouge-L, following [23, 6]. Consistent with Section 4.1, the threshold is set to be 0.5. With the same experimental setup, the results on the LLaMA-2-7b-chat model are shown in Table 5, where the effectiveness of our approach still holds.

| Model | Method | Single sampling | TRUTHFULQA | TYDIQA-GP |
|---|---|---|---|---|
| | Perplexity [38] | ✓ | 42.62 | 75.32 |
| | LN-Entropy [31] | ✗ | 44.77 | 73.90 |
| | Semantic Entropy [23] | ✗ | 47.01 | 71.27 |
| | Lexical Similarity [30] | ✗ | 67.78 | 45.63 |
| | EigenScore [6] | ✗ | 67.31 | 47.90 |
| LLaMA-2-7b | SelfCKGPT [32] | ✗ | 54.05 | 49.96 |
| | Verbalize [28] | ✓ | 53.71 | 55.29 |
| | Self-evaluation [21] | ✓ | 55.96 | 51.04 |
| | CCS [5] | ✓ | 59.07 | 71.62 |
| | CCS* [5] | ✓ | 60.12 | 77.35 |
| | HaloScope (OURS) | ✓ | **74.16** | **91.53** |

Table 5: **Main results with Rouge-L metric.** Comparison with competitive hallucination detection methods on different datasets. All values are percentages. "Single sampling" indicates whether the approach requires multiple generations during inference. **Bold** numbers are superior results.

# D Results with a Different Dataset Split

We verify the performance of our approach using a different random split of the dataset. Consistent with our main experiment, we randomly split 25% of the available QA pairs for testing using a different seed. HaloScope can achieve similar hallucination detection performance to the results in our main Table 1. For example, on the LLaMA-2-chat-7b model, our method achieves an AUROC of 76.39% and 94.89% on TRUTHFULQA and TYDIQA-GP datasets, respectively (Table 6). Meanwhile, HaloScope is able to outperform the baselines as well, which shows the statistical significance of our approach.

| Model | Method | Single sampling | TRUTHFULQA | TYDIQA-GP |
|---|---|---|---|---|
| | Perplexity [38] | ✓ | 56.71 | 79.39 |
| | LN-Entropy [31] | ✗ | 59.18 | 74.85 |
| | Semantic Entropy [23] | ✗ | 56.62 | 73.29 |
| | Lexical Similarity [30] | ✗ | 55.69 | 46.44 |
| | EigenScore [6] | ✗ | 47.40 | 45.87 |
| LLaMA-2-7b | SelfCKGPT [32] | ✗ | 55.53 | 51.03 |
| | Verbalize [28] | ✓ | 50.29 | 46.83 |
| | Self-evaluation [21] | ✓ | 56.81 | 54.06 |
| | CCS [5] | ✓ | 63.78 | 77.61 |
| | CCS* [5] | ✓ | 65.23 | 80.20 |
| | HaloScope (OURS) | ✓ | **76.39** | **94.98** |

Table 6: **Results with a different random split of the dataset.** Comparison with competitive hallucination detection methods on different datasets. All values are percentages. "Single sampling" indicates whether the approach requires multiple generations during inference. **Bold** numbers are superior results.

# E Ablation on Sampling Strategies

We evaluate the hallucination detection result when HaloScope identifies the hallucination subspace using LLM generations under different sampling strategies. In particular, our main results are obtained based on beam search, i.e., greedy sampling, which generates the next token based on the maximum likelihood. In addition, we compare with multinomial sampling with a temperature of

0.5. Specifically, we sample one answer for each question and extract their embeddings for subspace identification (Section 3.2), and then keep the truthfulness classifier training the same as in Section 3.3 for test-time hallucinations detection. The comparison in Table 7 shows similar performance between the two sampling strategies, with greedy sampling being slightly better.

| Unlabeled Data | TRUTHFULQA | TYDIQA-GP |
|---|---|---|
| Multinomial sampling | 76.62 | 93.68 |
| Greedy sampling (OURS) | **78.64** | **94.04** |

Table 7: Hallucination detection result under different sampling strategies. Results are based on the LLaMA-2-chat-7b model.

## F   Results with Less Unlabeled Data

In this section, we ablate on the effect of the number of unlabeled LLM generations $N$. Specifically, on TRUTHFULQA, we randomly sample 100-500 generations from the current unlabeled split of the dataset ($N$=512) with an interval of 100, where the corresponding experimental result on LLaMA-2-chat-7b model is presented in Table 8. We observe that the hallucination detection performance slightly degrades when $N$ decreases. Given that unlabeled data is easy and cheap to collect in practice, our results suggest that it's more desirable to leverage a sufficiently large sample size.

| $N$ | TRUTHFULQA |
|---|---|
| 100 | 73.34 |
| 200 | 76.09 |
| 300 | 75.61 |
| 400 | 73.00 |
| 500 | 75.50 |
| 512 | **78.64** |

Table 8: The number of the LLM generations and its effect on the hallucination detection result.

## G   Results of Using Other Uncertainty Scores for Filtering

We compare our HaloScope with training the truthfulness classifier by membership estimation with other uncertainty estimation scores. We follow the same setting as HaloScope and select the threshold $T$ and other key hyperparameters using the same validation set. The comparison is shown in Table 9, where the stronger performance of HaloScope vs. using other uncertainty scores for training can precisely highlight the benefits of our membership estimation approach by the hallucination subspace. The model we use is LLaMA-2-chat-7b.

| Method | TRUTHFULQA | TYDIQA-GP |
|---|---|---|
| Semantic Entropy | 65.98 | 77.06 |
| SelfCKGPT | 57.38 | 52.47 |
| CCS* | 69.13 | 82.83 |
| HaloScope (OURS) | **78.64** | **94.04** |

Table 9: Hallucination detection results leveraging other uncertainty scores.

## H   Results on Additional Tasks

We evaluate our approach on two additional tasks, which are (1) text continuation and (2) text summarization tasks. For text continuation, following [32], we use LLM-generated articles for a specific concept from the WikiBio dataset. We evaluate under the sentence-level hallucination detection task and split the entire 1,908 sentences in a 3:1 ratio for unlabeled generations and test data. (The other implementation details are the same as in our main Table 1.)

For text summarization, we sample 1,000 entries from the HaluEval [25] dataset (summarization track) and split them in a 3:1 ratio for unlabeled generations and test data. We prompt the LLM with "[document] \n Please summarize the above article concisely. A:" and record the generations while keeping the other implementation details the same as the text continuation task.

The comparison on LLaMA-2-7b with three representative baselines is shown below. We found that the advantage of leveraging unlabeled LLM generations for hallucination detection still holds.

| Method | Text continuation | Text summarization |
|---|---|---|
| Semantic Entropy | 69.88 | 60.15 |
| SelfCKGPT | 73.23 | 69.91 |
| CCS$^*$ | 76.79 | 71.36 |
| HaloScope (OURS) | **79.37** | **75.84** |

Table 10: Hallucination detection results on different tasks.

## I  Broader Impact and Limitations

**Broader Impact.** Large language models (LLMs) have undeniably become a prevalent tool in both academic and industrial settings, and ensuring trust in LLM-generated content for safe usage has emerged as a paramount concern. In this line of thought, our paper offers a novel approach HaloScope to detect LLM hallucinations by leveraging the in-the-wild unlabeled data. Given the simplicity and versatility of our methodology, we expect our work to have a positive impact on the AI safety domain, and envision its potential usage in industry settings. For instance, within the chat-based platforms, the service providers could seamlessly integrate HaloScope to automatically examine the factuality of the LLM generations before information delivery to users. Such applications will enhance the reliability of AI systems in the current foundation model era.

**Limitations.** Our new algorithmic framework aims to detect LLM hallucinations by harnessing the unlabeled LLM generations in the open world, and works by devising a scoring function in the representation subspace for estimating the membership of the unlabeled instances. While HaloScope offers a straightforward solution to leveraging the unlabeled data for training, its effectiveness is still somewhat affected by the drastic distribution shift between the unlabeled data and the test data. Therefore, a distributionally robust algorithm for training the hallucination classifier is a promising future work.

## J  Software and Hardware

We run all experiments with Python 3.8.5 and PyTorch 1.13.1, using NVIDIA RTX A6000 GPUs.

