# OpenReview forum: "HaloScope: Harnessing Unlabeled LLM Generations for Hallucination Detection"
_NeurIPS.cc/2024/Conference — NeurIPS 2024 spotlight_

### Official Review · Reviewer_JMXE · 2024-07-08

**Soundness:** 3
**Presentation:** 3
**Contribution:** 3
**Rating:** 6
**Confidence:** 5

**Summary:**

This paper proposes a new way of harnessing unlabeled LLM generation as the training set for fact verification. The key assumption is "LLM generates factually correct statements more than hallucinogenic statements", which guarantees the existence of clear subspace between hallucination and non-hallucination statements. As the mean of identifying such subspace, singular decomposition is applied to find out orthonormal vector to project LLM's embeddings. The distance from origin is regarded as the value to define "abnormality". The experiments demonstrate that the proposed idea works out and outperforms several existing methods.

**Strengths:**

* The idea of using unlabeled LLM generation is interesting and show the potential over other counter parts.
* Ablation study may include key components we can alter.
* The paper is clearly written and solve an important challenge - distinguishing hallucination or not.

**Weaknesses:**

There are many weaknesses and things to be considered for further improvements.

**Limited Training Set:** The main advantage of the propose method is the usage of unlabeled LLM generation. It means, we can create much larger training set by generating diverse QA pairs (including multi-domain, subjects, etc). But this study is limited to using only the give training set from a few data sets. Also, the classification performance is much lower than what we can obtain by using labeled set in Figure 5. Showing the potential how this framework gets synergies by adding more LLM generations. (we can simply create such large data by using LLMs with simple prompts or we can add LLM generations on other large-scale QA bench not for fact check). Can the fact verification performance improve by adding more LLM generations?

**Issue on Robustness Comparison:** I don't understand why the proposed method gets higher robustness compared with other methods. The propose one still relies on training on a specific data. How could this increase robustness over others? The latent space extracted by a specific dataset cannot be generalized to all domains. I think this is the similarity between the two datasets, TRIVIA QA, TrustfulQA. Please clearly explain what data distribution, domain difference the two data have, and what components help the robustness against distribution shift.

**Explanation on Hyperparameter:** In Lines 150-151, I think the methods need $T$ to split the LLM generations into two different groups of $\mathcal{H}$ and $\mathcal{T}$. But, there is no mention about this, and any other studies on ablating the value of $T$.

**Visualization Study**: A key assumption is the existence of clear subspace to distinguish hallu and non-hallu statements. I think that the author should provide some qualitative analysis using visualization. We can visualize the projected embedding space with oracle labels on 2D space. This clearly show that how such two classes are separable well in latent space.

**Questions:**

No Questions.

**Limitations:**

Yes, author stated the limitation on Page 17.

---

> ### Author Rebuttal · Authors · 2024-08-04
>
> We thank you for recognizing our work as interesting and for studying an important challenge. We appreciate the reviewer's comments and suggestions, which we address below:
>
>
>
> **A1. The effect of adding more unlabeled data**
>
> Thank you for the suggestion! Our ablation on the number of unlabeled data for hallucination detection in **Appendix Table 8** indeed shows that scaling up the unlabeled data is beneficial. To further verify this, as suggested, on the TruthfulQA dataset, we have explored (1) adding LLM-generated sentences for a specific concept from the WikiBio dataset [1] into the unlabeled data and (2) adding LLM generations for a different TriviaQA dataset into the unlabeled data. The hallucination detection results with different numbers of added samples are shown as follows (the model is LLaMA-2-7b and the test dataset is TruthfulQA):
>
>
>
> |     Added samples   | WikiBio |  Added samples | TriviaQA|
> | ------ | ----- | ----- | ----- |
> |0 |78.64  |0 |  78.64|
> |400 | 79.27|2000 | 80.81 |
> |800 | 79.94| 4000| 83.37|
> |1200 |80.90 | 6000|82.96 |
> |1600 | 81.66|8000|84.19 |
>
>
> We observe a similar trend as in Table 8 of our paper and will be happy to add more discussion on this in the revised version.
>
>
>
> **A2. Discussion on robustness comparison**
>
>
> Thank you for pointing this out! Firstly, we concur with the reviewer's opinion that our approach can be affected by the distribution shift between the unlabeled data and the test data, which we also discuss in the Limitation section of our paper.
>
>
>
> *Although we did not explicitly claim the better robustness of our approach compared to other alternatives*, we are happy to clarify the method robustness and its relationship with dataset similarity. Specifically, the four datasets used in our paper have the following domain differences:
>
>
>
>
> - TruthfulQA: This dataset includes questions from various domains such as health, law, fiction, conspiracies, etc. It is designed to measure the truthfulness of language models by including questions that some humans might answer falsely due to misconceptions.
> - TriviaQA: This dataset includes question-answer pairs from trivia enthusiasts. The questions cover a wide range of topics and are paired with evidence documents from sources like Wikipedia and web search results.
> - CoQA: This dataset includes passages from seven diverse domains: children’s stories, literature, middle and high school English exams, news, Wikipedia, Reddit, and science.
> - TyDiQA: This dataset consists of questions related to Wikipedia articles.
>
>
> Based on this and Figure 3(a) of our paper, we have made the following observations:
>
> **The knowledge scope matters for method transferability**: We find that a hallucination subspace calculated within a more general knowledge scope can transfer well to data with a smaller knowledge scope, but not vice versa. For example, the subspace extracted from TruthfulQA (where the knowledge scope is more restricted compared to other datasets) demonstrates less generalization ability to other datasets. The detection AUROC drops from 76.42% to 67.81% when the TruthfulQA subspace is used for hallucination detection on the CoQA test set, while the CoQA subspace achieves an AUROC of 77.36% on the TruthfulQA test set (only a 1.28% decrease). Additionally, our approach works well when the unlabeled data and the test data have a similar knowledge scope. For instance, the TriviaQA and CoQA datasets have similar knowledge scopes (Wikipedia and other web knowledge), allowing the subspace learned from one dataset to generalize to the other, as verified by our experiments.
>
> We believe this explains why, in certain cases, our approach is robust when transferring to other data distributions. We would be happy to add this discussion to our revised paper.
>
> **A3. Explanation on the hyperparameter**
>
> Great point. As explained in **line 203** of our paper, we determine the threshold $T$ on a separate validation set consisting of 100 LLM generations. For the LLaMA-2-7b model with the TruthfulQA dataset, we provide further ablation results of $T$ on the test set as follows. Here, "max" and "min" denote the maximal and minimal membership scores of the unlabeled data.
>
> |     T  | AUROC (in %)|
> | ------ | ----- |
> |(max - min) * 10% + min | 69.58|
> |(max - min) * 20% + min | 75.96|
> |(max - min) * 30% + min | 73.20|
> |(max - min) * 40% + min | 77.37|
> |(max - min) * 50% + min | 77.43|
> |(max - min) * 60% + min | 79.19|
> |(max - min) * 70% + min | 78.64|
> |(max - min) * 80% + min | 74.67|
> |(max - min) * 90% + min | 70.31|
>
> The final reported result (78.64%) in our main Table 1 is selected based on the AUROC on the validation set.
>
>
> **A4. Visualization**
>
>
> Another great point. As suggested, we provide visualization results [here](https://openreview.net/attachment?id=ukMLqTlT90&name=pdf) on the embeddings of the truthful and hallucinated LLM generations for the TyDiQA-GP dataset with the LLaMA-2-7b model. These embeddings are almost linearly separable and align well with empirical observations in the literature [2].
>
>
> [1] Manakul et al., SELFCHECKGPT: Zero-Resource Black-Box Hallucination Detection for Generative Large Language Models, EMNLP 2023
>
> [2] Zou et al., Representation engineering: A top-down approach to ai transparency. arXiv preprint arXiv:2310.01405, 2023.

---

> > ### Comment · Reviewer_JMXE · 2024-08-09
> > **Response to Authors**
> >
> > Thanks for the clear response on my concerns and questions. All the things has been resolved, so I increase my score to 6. Thanks!

---

> > > ### Author Response · Authors · 2024-08-09
> > > **Thank you!**
> > >
> > > Thank you so much for taking the time to read our response and increase your rating! We are glad that our rebuttal addresses your concerns.
> > >
> > > Thanks,
> > > Authors

---

### Official Review · Reviewer_tABH · 2024-07-13

**Soundness:** 4
**Presentation:** 4
**Contribution:** 4
**Rating:** 9
**Confidence:** 4

**Summary:**

The paper attempts to address the popular problem of hallucination in texts generated by today's LLMs (large language models). Hallucination refers to false or misleading text generated by LLMs. The paper attempts to address hallucination by proposing a truthfulness classifier, HaloScope, that operates on the unlabeled text generated by an LLM during operation. The paper compares Haloscope with existing algorithms and shows its superior performance.

The key idea behind Haloscope is identifying a hallucination subspace in the space of latent (Euclidean) representations of an LLM generation. Thus, when when the latent representation of a generation aligns strongly with this hallucination subspace, the text is classified as potentially hallucinated.

**Strengths:**

The paper is very well-written. The key idea behind HaloScope while being simple in retrospect is novel and a welcome contribution. The exposition in the paper is very clear and the simulations have been excellently presented with all the relevant details. The work on ablation studies (Sections 4.2-4.4) is also a welcome contribution and adds to the overall value of the work.

The reviewer believes the research community will benefit and draw upon the ideas and simulations presented in this paper.

**Weaknesses:**

One point I would like to note is with regard to the problem setup. Currently, the way problem is formulated (Equations 1, 2, 3, 4), it appears that the fact that an LLM generation is a hallucination is independent of the corresponding user prompt. This does not seem to be a good assumption to make. However, the empirical results do outweigh the weakness of such an assumption. The reviewer suggests the authors to describe their modeling choice (Eq 4) a little more in detail, specially how it relates to the user-prompt conditioning.

**Questions:**

Line 40-41. By assuming the specific mixture of two distributions. The authors are assuming that hallucinated data is generated with probability \pi independent of the input. Can authors please elaborate.

Line 158. What is the difference between lambda and capital T?

**Limitations:**

The distribution shift has been identified as a potential barrier to a reliable usage of HaloScope by the authors. The reviewer agrees with the authors and appreciate their candidness. The reviewer would like to suggest mentioning this as potential future work direction in the conclusion.

---

> ### Author Rebuttal · Authors · 2024-08-04
>
> We are deeply encouraged that you recognize our method to be novel, welcome, and beneficial to the research community.
>
> Your summary and comments are insightful and spot-on :)
>
>
> **A1. Clarification on the problem setup**
>
> You raise a great point! We agree with you that the truthfulness of an LLM generation should be dependent on the given user prompt. We plan to revise it as follows: define a new variable $\widehat{\mathbf{x}}$ that denotes the concatenation of the user prompt and the LLM generation, and the distribution P_unlabeled, P_true, P_hal are thus defined over the new variable. Moreover, that will also address your confusion on the assumption that hallucinated data is generated with probability $\pi$ independent of the input. Thank you for catching this issue!
>
>
>
>
>
> **A2. Clarification on the thresholds**
>
> Thank you for the question! We are happy to clarify their differences. $T$ is the threshold when determining the membership of the unlabeled data (true or false) using the membership estimation score (Equation 7 of our paper), which can be chosen on a small amount of validation data as discussed in Section 4.1 of the paper. $\lambda$ is the threshold on the probabilistic outputs from the hallucination classifier during test time to determine whether a testing LLM generation is hallucinated or not.
>
>
>
> **A3. Mention distribution shift in the conclusion**
>
> Certainly! We will be happy to discuss this future work in the conclusion section. Possible extension ideas can include firstly setting up new experimental environments where the unlabeled data and the test data belong to different domains (such as between daily dialogues and medical question-answer pairs), and then developing a distributionally robust algorithm for training. Another interesting idea to explore might be out-of-sample extension for SVD [1] that specifically deals with the reconstruction and projection precision for samples that are not in the training set.
>
>
>
> [1] Bengio et al., Out-of-Sample Extensions for LLE, Isomap, MDS, Eigenmaps, and Spectral Clustering, NIPS 2003

---

> > ### Comment · Reviewer_tABH · 2024-08-09
> >
> > Thank you for the responses.
> > A1. Could you write down the new equations that will result so I can get a clear picture?
> > A2. Acknowledged.
> > A3. Acknowledged.

---

> > > ### Author Response · Authors · 2024-08-09
> > > **Author response**
> > >
> > > Thank you for taking the time to read our rebuttal! We are happy to further respond to your question on A1, with revised definitions of hallucination detection and unlabeled data.
> > >
> > > ---------
> > > We first generate the output $x$ conditioned on the prompt $x_\text{prompt}$. Both the prompt and generation will be used in subsequent hallucination detection, which is defined as follows.
> > >
> > > **Definition 2.2 (Hallucination detection)**
> > > We denote $P_\text{true}$ as the joint distribution over the truthful input and generation pairs, which is referred to as truthful distribution. For any given generated text $x$ and its corresponding input prompt $x_\text{prompt}$ where $(x_\text{prompt}, x) \in \mathcal{X}$, the goal of hallucination detection is to learn a binary predictor $G: \mathcal{X} \rightarrow \\{0,1\\}$ such that
> > > \begin{equation}
> > >     G({x_\text{prompt}, x}) = \begin{cases}
> > >         1, &\text{if }  {(x_\text{prompt}, x)} \sim P_\text{true} \\\\
> > >         0,         &otherwise
> > >     \end{cases}
> > > \end{equation}
> > >
> > > **Definition 3.1 (Unlabeled data distribution)** We define the unlabeled LLM input and generation pairs to be the following mixture of distributions
> > > \begin{equation}
> > >     P_{\text{unlabeled}} = (1-\pi) P_{\text{true}} + \pi P_{\text{hal}},
> > > \end{equation}
> > >  where $\pi \in (0,1]$. Note that the case $\pi = 0$ is idealistic since no false information occurs. In practice, $\pi$ can be a moderately small value when most of the generations remain truthful.
> > >
> > >  **Definition 3.2 (Empirical dataset)** An empirical set $\mathcal{M} = \\{(x_{\text{prompt}}^1, x_1), ..., (x_{\text{prompt}}^N, x_N)\\}$ is sampled independently and identically distributed (i.i.d.) from this mixture distribution $P_{\text{unlabeled}}$, where $N$ is the number of samples. $x_i$ denotes the response generated with respect to some input prompt $x_{\text{prompt}}^i$.
> > >
> > > Note that in implementation, we indeed consider passing both the prompt and the LLM-generated answer to obtain the embedding, which aligns with our problem definitions. These updates have been made accordingly in our manuscript. Thank you again for the helpful suggestion!

---

> > > > ### Comment · Reviewer_tABH · 2024-08-11
> > > >
> > > > Thank you for writing down the equations. I acknowledge the response.

---

> > > > > ### Author Response · Authors · 2024-08-11
> > > > > **Thank you!**
> > > > >
> > > > > Thank you for acknowledging our response and the equations. We're glad to have provided clarity. If you have any additional questions or feedback, please feel free to reach out!

---

### Official Review · Reviewer_6VqW · 2024-07-13

**Soundness:** 3
**Presentation:** 3
**Contribution:** 3
**Rating:** 6
**Confidence:** 3

**Summary:**

This paper presents a technique for detecting hallucinations by leveraging unlabeled data generation. Instead of relying on human annotation, the method automatically distinguishes between truthful and untruthful generations using network embeddings and their projection onto singular vectors with high singular values. Subsequently, this information is utilized to train a classifier. The study demonstrates enhancements across multiple benchmarks compared to several baseline methods.

**Strengths:**

* Leveraging unlabeled generation for hallucination detection without any need for human annotation.
* The method scales well for larger models.
* The method does not require sampling multiple generations.

**Weaknesses:**

* The method relies on BLUERT for ground truth evaluation. Can you justify the use of BLUERT score?

**Questions:**

* Have you explored the effectiveness of the method on tasks beyond QA, such as summarization?
* Can you provide more details on the result of Table 3? What data is used to perform this ablation? This is important because where to extract the embeddings has huge effect on the effectiveness of the method.

**Limitations:**

Yes

---

> ### Author Rebuttal · Authors · 2024-08-04
>
> We are glad to see that the reviewer recognized the strengths of our work from various perspectives. We thank the reviewer for the thorough comments and suggestions. We are happy to clarify as follows:
>
>
> **A1. Clarification on the BLEURT metric**
>
> Thank you for pointing this out! BLEURT [1] is designed to evaluate the quality of text by comparing it to reference texts, similar to traditional metrics like BLEU. However, BLEURT leverages pretrained Transformer models, such as BERT, to capture deeper semantic nuances and contextual meanings. This makes it particularly effective for detecting subtle discrepancies between generated outputs and reference ground truths, which is critical in identifying hallucinations that may not be immediately obvious through surface-level evaluation.
>
> In addition, hallucinations can vary widely in form, from subtle factual inaccuracies to more blatant falsehoods. BLEURT's embedding-based approach allows it to handle this variability better than traditional n-gram-based metrics, which might miss nuanced errors. This robustness ensures that the evaluation can effectively capture both major and minor hallucinations, providing a more comprehensive assessment of the model's output quality.
>
> Finally, BLEURT has been shown to correlate well with human judgments in various natural language generation tasks [1, 2], which makes it a reliable proxy for assessing the factual correctness and coherence of LLM outputs, which is essential in hallucination detection. **Additionally, we show that the effectiveness of our algorithm is robust under a different similarity measure, ROUGE, in Appendix D**, which is based on substring matching.
>
>
>
>
> **A2. Effectiveness of our approach on additional tasks**
>
> Thank you for the suggestion! We evaluate our approach on two additional tasks, which are (1) text continuation and (2) text summarization tasks.
>
> For text continuation, following [1], we use LLM-generated articles for a specific concept from the WikiBio dataset. We evaluate under the sentence-level hallucination detection task and split the entire 1,908 sentences in a 3:1 ratio for unlabeled generations and test data. (The other implementation details are the same as in our original submission.)
>
> For text summarization, we sample 1,000 entries from the HaluEval [3] dataset (summarization track) and split them in a 3:1 ratio for unlabeled generations and test data. We prompt the LLM with "[document] \n Please summarize the above article concisely. A:" and record the generations while keeping the other implementation details the same as the text continuation task.
>
>
> The comparison on LLaMA-2-7b with three representative baselines is shown below. We found that the advantage of leveraging unlabeled LLM generations for hallucination detection on Wikipedia articles still holds.
>
>
>
> |     Method   | Text continuation | Text summarization|
> | ------ | ----- | ----- |
> |Semantic Entropy |69.88|60.15 |
> |SelfCKGPT | 73.23|69.91 |
> |CCS∗ | 76.79| 71.36|
> |HaloScope (Ours) | **79.37**|**75.84**|
>
>
> **A3. Details of Table 3**
>
> Absolutely! As in the first row of Table 3, we use 4 datasets (TruthfulQA, TriviaQA, CoQA, and TyDiQA-GP) to ablate on the effect of where the embedding is extracted for hallucination detection. The results are the detection AUROCs based on different locations in a typical transformer block to extract the embeddings, i.e., the output of each transformer block, the output of the self-attention module, and the output of the MLP feedforward layer. From Table 3, we observe that the LLaMA model tends to encode the hallucination information mostly in the output of the transformer block, while the most effective location for OPT models is the output of the feedforward layer.
>
> [1] Sellam et al., BLEURT: Learning Robust Metrics for Text Generation, ACL 2020.
>
> [2] Bubeck et al., Sparks of Artificial General Intelligence: Early experiments with GPT-4, arXiv preprint, 2303.12712.
>
> [3] Li et al., HaluEval: A Large-Scale Hallucination Evaluation Benchmark for Large Language Models, EMNLP 2023.

---

> > ### Comment · Reviewer_6VqW · 2024-08-11
> >
> > I appreciate the authors for addressing my concerns.

---

> > > ### Author Response · Authors · 2024-08-11
> > > **Thank you!**
> > >
> > > Thank you for your appreciation. We're glad we could address your concerns! Please let us know if you have any further questions or suggestions.

---

### Official Review · Reviewer_smZn · 2024-07-15

**Soundness:** 3
**Presentation:** 3
**Contribution:** 4
**Rating:** 8
**Confidence:** 4

**Summary:**

The paper is very clear in the problem it is facing and the solution it proposes is also clearly described. Specifically it is looking at detecting hallucinations produced by generative language models. It does so by taking internal representations of the LLM, projecting these onto SVD factorisation which identifies important directions of variation for the models subspace(s) over some relevant data.

Membership of unlabelled data (hallucination, or not) is then obtained by projecting the centred representation for the datapoint at hand against the principle singular vector. This provides an unsupervised labelling method. This is used to label all data available, and then a binary classifier is trained on this. This is the final hallucination classifier.

Experimental results are given over a few Q+A datasets and against several related baselines for hallucination detection. The proposed method is shown to be the most accurate consistently.

It's actually rather amazing I think that the projection against the primary SVD vector (with scores aggregated across these from different parts of the network) works so well as an unsupervised labeller of truthfullness/hallucination. There's no specific information here at all about the problem of hallucination, and this could have captured the information of any other trait (or even meant nothing at all).

**Strengths:**

* Very interesting empirical results on the provided Q+A datasets.
* Method is clear and simple, assuming access available to the internals of the LLM.
* The ablations across all of section 4.3 are very thorough. These answered the questions I had noted down reading up to that point of the paper.

**Weaknesses:**

* Hallucinations do vary a lot depending on the specific problem at hand, e.g. in question-answering with or without a prompt, or in data-to-text NLG. This paper only looks at question-answer based tasks. It would be interesting to see the method applied on other tasks. There is nothing about the proposed method which limits where it can be applied, it's purely an empirical question as to whether the results are the similar or rather different.

* Access to the internals of proprietary, cloud hosted LLMs is not a given. Often only the predictions are obtainable for these. The method won't be applicable in this case.

**Questions:**

* Why is projecting against these subspace primary vectors so informative for hallucination detection? I'm quite surprised at how well this works, given that could capture any other feature of the model/data other than hallucination. The variation of this across the different network layers is significant as shown in Figure 3 c. Based on that, the practical way to apply this would be to measure on every layer before selecting which to use right?

* Would it be possible/sensible/silly to incorporate an objective on certain subspaces into the training of the model (if not just zero-shoting the task of interest with an already capable LLM)?

**Limitations:**

* Access to the internals of the LLM is the main one, as already noted.
* More empirical results over different types of language generation tasks which each have their own nuances to what is a hallucination would be of interest.

---

> ### Author Rebuttal · Authors · 2024-08-04
>
> We thank the reviewer for the comments and suggestions. We are encouraged that you recognize our approach to be clear and with interesting experiments and thorough ablations. We address your questions below:
>
> **A1. Effectiveness of our approach on additional tasks**
>
> Thank you for the suggestion! We evaluate our approach on two additional tasks, which are (1) text continuation and (2) text summarization tasks.
>
> For text continuation, following [1], we use LLM-generated articles for a specific concept from the WikiBio dataset. We evaluate under the sentence-level hallucination detection task and split the entire 1,908 sentences in a 3:1 ratio for unlabeled generations and test data. (The other implementation details are the same as in our original submission.)
>
> For text summarization, we sample 1,000 entries from the HaluEval [2] dataset (summarization track) and split them in a 3:1 ratio for unlabeled generations and test data. We prompt the LLM with "[document] \n Please summarize the above article concisely. A:" and record the generations while keeping the other implementation details the same as the text continuation task.
>
> The comparison on LLaMA-2-7b with three representative baselines is shown below. We found that the advantage of leveraging unlabeled LLM generations for hallucination detection still holds.
>
> |     Method   | Text continuation | Text summarization|
> | ------ | ----- | ----- |
> |Semantic Entropy |69.88|60.15 |
> |SelfCKGPT | 73.23|69.91 |
> |CCS∗ | 76.79| 71.36|
> |HaloScope (Ours) | **79.37**|**75.84**|
>
> **A2. Discussion on access to internals of proprietary, cloud hosted LLMs**
>
> You raise a great point. We concur with your opinion that access to the internal representations is not easy for proprietary, cloud-hosted LLMs. We provide our understanding on this as follows:
>
> - Firstly, we believe that hallucination detection is still an ongoing research topic, and the research efforts on exploring the internal representations of open-sourced models are equally, if not more, important than the research assuming black-box access to LLMs. Such access to model internals is beneficial for transparency and debugging, which can help us understand where, when, and how the hallucinated generations occur. By investigating specific layers or components where the model's reasoning deviates, researchers and developers can debug and refine the model more effectively, leading to improvements in both performance and safety.
> - Secondly, internal representations capture the nuanced, multi-layered information that LLMs process as they generate responses compared to the textual outputs. By analyzing these representations, we gain access to a more detailed understanding of the model's internal decision-making process. This granularity allows us to identify potential hallucination subspaces, leading to better hallucination detection performance compared to existing black-box approaches, such as SelfCKGPT [1], Self-evaluation [3], etc., which are already compared in our submission (**Table 1**).
> - Finally, we may consider some strategies to mitigate the concerns about the proprietary nature of cloud-hosted LLMs, such as experimenting with proxy models that closely approximate the behavior of the proprietary LLMs. This can help study and detect hallucinations without infringing on intellectual property rights.
>
> Thank you for bringing this up with us! We will include the discussion in the Limitation section of our submission.
>
> **A3. Discussion on subspace primary vectors and the layer-wise variations**
>
> You raise a great point!
>
> Firstly, subspace primary vectors derived through SVD often encapsulate the dominant patterns and variations within the internal representations of a model [4]. These vectors can highlight the primary modes of variance in the unlabeled data, which are not purely random but instead capture significant structural features of the model’s processing. In our case, it could be the hallucination information. Even though these vectors could, in theory, capture various features, they are particularly informative for detecting hallucinations because hallucination and truthfulness patterns are among these primary modes of variation in the unlabeled data. This phenomenon can be verified by the empirically observed separability in both our submission (**Figure 7 in Appendix**) and literature [5]. In addition, we provide visualization results [here](https://openreview.net/attachment?id=ukMLqTlT90&name=pdf) on the embeddings of the truthful and hallucinated LLM generations for the TyDiQA-GP dataset on the LLaMA-2-7b model, which are almost linearly separable. We believe that this can help illustrate the effectiveness of projection against the top singular vectors.
>
> Moreover, for the variation with respect to the layers, we select the layer based on a small number of validation data as described in **Section 4.1** of our submission.
>
> **A4. Incorporating training objective on certain subspaces**
>
> Another great point! We do anticipate the possibility of explicitly regularizing the training of LLMs for better hallucination detection. One straightforward idea is to add an objective that fine-tunes the LLMs (or their feature subspace) to clearly distinguish the true vs. false based on the representation space and the separated unlabeled data at each training step, and then keep training for a few epochs. This could be a promising future work.
>
> [1] Manakul et al., SELFCHECKGPT: Zero-Resource Black-Box Hallucination Detection for Generative Large Language Models, EMNLP 2023
>
> [2] Li et al., HaluEval: A Large-Scale Hallucination Evaluation Benchmark for Large Language Models, EMNLP 2023.
>
> [3] Kadavath et al., Language models (mostly) know what they know. arXiv preprint arXiv:2207.05221, 2022.
>
> [4] https://en.wikipedia.org/wiki/Principal_component_analysis
>
> [5] Zou et al., Representation engineering: A top-down approach to ai transparency. arXiv preprint arXiv:2310.01405, 2023

---

### Author Rebuttal · Authors · 2024-08-04

We thank all the reviewers for their time and valuable comments. We are encouraged to see that all reviewers find our approach **interesting, clear, simple, new, welcome**, and **scales well** (smZn, 6VqW, tABH, JMXE), and our results **very interesting, excellently presented**, with **thorough, welcome** ablations (smZn, tABH). Reviewers also recognize our paper presentation to be **clear, very well-written** (smZn, tABH, JMXE).

As recognized by multiple reviewers, the significance of our work can be summarized as follows:


- Our work offers a new algorithmic framework that leverages the unlabeled LLM generations to help hallucination detection, which is an important research question.
- The framework is based on the factorization of the LLM representations, where the membership of the unlabeled data is inferred and subsequently, a final hallucination classifier is learned. The approach is simple, clear, and effective.
- We provide supportive experiments to show the effectiveness of our approach, precisely highlighting how the proposed framework works in practice. Sufficient ablations are provided to help readers understand the method.

We respond to each reviewer's comments in detail below. We are happy to revise the manuscript according to the reviewers' suggestions, and we believe this will make our paper stronger.

---

### Decision · Program_Chairs · 2024-09-25

**Decision:**

Accept (spotlight)

**Comment:**

This paper receives all positive recommendations. Reviewers agree that the paper is very novel and well-written and has a significant impact in the community. Minor comments are well addressed during the rebuttal. The AC agrees with the positive opinion and recommends acceptance.